# Analytically Pricing Formula for Contingent Claim with Polynomial Payoff under ECIR Process

Fukiat Nualsri and Khamron Mekchay *

Department of Mathematics and Computer Science, Faculty of Science, Chulalongkorn University, Bangkok 10330, Thailand; 6373011023@student.chula.ac.th
* Correspondence: khamron.m@chula.ac.th

**Abstract:** Contingent claims, such as bonds, swaps, and options, are financial derivatives whose payoffs depend on uncertain future real values of underlying assets which emphasize various real-world applications. In general, valuations for contingent claims can be derived from the conditional expectations of underlying assets. For a simple process, the moments are usually directly obtained from its transition probability density function (PDF). However, if the transition PDF is unavailable in simple form, the derivations of the moments and the contingent claim prices will not be accessible in closed forms. This paper provides a closed-form formula for pricing contingent claims with nonlinear payoff under a no-arbitrage principle when underlying assets follow the extended Cox–Ingersoll–Ross (ECIR) process with the symmetry properties of the Brownian motion. The formula proposed here is a consequence of successfully solving an explicit solution for a system of recurrence partial differential equations in which its solution subtly depends on the conditional moments. For the particular CIR process, we obtain simple closed-form formulas by solving the Riccati differential equation. Furthermore, we carry out a complete investigation of the convergent case for those formulas. In case such as that of the unsolvable Riccati differential equation, ECIR case, a numerical method for numerically evaluating the mentioned analytical formulas and numerical validations for the formulas are examined. The validity and efficiency of the formulas are numerically demonstrated by comparison with results from Monte Carlo simulations for various modeling parameters. Finally, the proposed formula is applied to the value contingent claims such as coupon bonds, interest rate swaps, and arrears swaps.

**Keywords:** closed-form formula; conditional expectation; contingent claim; ECIR process

**MSC:** 60G65; 91G20

## 1. Introduction

Conditional expectations are useful statistical values applied in many branches in science, especially in describing behaviors of observed data, and are often studied from a probabilistic viewpoint based on transition probability density functions (PDFs) of the data. Many applications in finance and economics require the knowledge of conditional expectations, for example, in the valuation of financial products, e.g., contingent claims such as coupon bonds, swaps, discount factor, etc., as can be seen in the works of Ben-Ameur et al. [1] and Grasselli [2] for more details.

To value financial products, the no-arbitrage principle is essential; if arbitrage exists, it is guaranteed that the investor can make profit from nothing, which provides an investment opportunity with infinite return. To satisfy the arbitrage-free property, symmetric information is required [3]. The valuation of a contingent claim is usually investigated

based on the conditional expectation (1) under a filtered risk-neutral probability space $(\Omega, \mathcal{F}_t, \{\mathcal{F}_t\}_{0 \leq t \leq T}, Q)$ in the form

$$\mathbb{E}^Q \left[ e^{-\int_t^T (\alpha x_s + \beta)\, ds} f_T + \int_t^T e^{-\int_t^s (\alpha x_u + \beta)\, du} g_s\, ds \mid \mathcal{F}_t \right], \tag{1}$$

where $\alpha, \beta \in \mathbb{R}$ and $\{x_t\}_{0 \leq t \leq T}$ is an adapted stochastic process describing the underlying asset, and for some real value functions $f_T$ and $g_s$ with $0 \leq t < s \leq T$.

In this work, we consider the conditional expectations (1) in the form

$$\mathbb{E}^Q \left[ x_T^\gamma e^{-\int_t^T (\alpha x_s + \beta)\, ds} \mid \mathcal{F}_t \right] = \mathbb{E}^Q \left[ x_T^\gamma e^{-\int_t^T (\alpha x_s + \beta)\, ds} \mid x_t = x \right], \tag{2}$$

where $\gamma \in \mathbb{R}$ and to be more specific, the process $x_t$ is considered as a short rate described by the extended Cox–Ingersoll–Ross (ECIR) process [4]. The case that $\gamma = 1$ in (2) was also studied in Duffie et al.'s work [5] for the class of affine jump-diffusion processes with generalized payoff, including some case studies such as those of Ornstein–Uhlenbeck (OU) and the Cox–Ingersoll–Ross (CIR) or squared processes. To be more general, the Wiener process $W_t$ in the ECIR process can be generalized by using the mixed fractional Brownian motion, i.e., a linear combination of the Wiener process $W_t$ and the fractional Brownian motion $W_t^H$ with the Hurst parameter, $H \in (0, 1)$. The mixed fractional Brownian motion has some useful characteristic, e.g., it is arbitrage free and gains more attention for applications in finance. An example of an application based on the mixed fractional Brownian motion is described in [6].

Note that the contingent claim prices in the forms (1) and (2) have many useful applications. In finance, a non-arbitrage price at time $t$ of financial derivatives is considered on a conditional expectation under a risk-neutral measure of their discounted payoff, more details of which can be found in [7]. Therefore, the valuations of financial derivatives, such as the coupon bonds, variance swaps, interest rate swaps, and options, always involve calculating the forms of conditional expectations (1) and (2).

Since many processes describing financial assets have no transition PDFs in simple form, the conditional expectations, the valuation of many financial products involving (1) and (2) usually are not accessible in closed form; thus, alternative methods are required. In practice, when analytical formulas for the expectations are not known in concise form, a practical method such as the Monte Carlo simulation is required, which has disadvantages in term of computational time. This paper aims to propose a closed-form formula for the conditional expectation (1) based on the solution of a partial differential equation (PDE) according to the Feynman–Kac representation, without requiring the knowledge of the transition PDF. In some applications, the related PDEs may have no closed form solution and numerical methods are required in order to obtain the results, for example, Ahmadian and Ballestra [8] proposed the finite element method to solve ruin-related problems, and Liang and Zou [9] studied the valuation of credit contingent interest rate swap with credit rating migration using the alternating direction implicit method.

The CIR process is a diffusion process satisfying the Pearson Equation [10] and involving a wide variety of issues in many branches—more details on this can be found in [11]. This process was initially introduced by Feller [12] as a population growth stochastic model and becomes popular in finance when Cox et al. [13] applied it to describe the evolution behavior of short-term interest rates. Even though the CIR process is very useful in terms of pricing financial derivatives, especially short-term interest rates, the process has a limitation on its constant parameters, which are not suitable for modeling time-varying observed data. A lot of strong empirical evidence has found that extreme movements in finance-based practices tend to be assumed in function of the time, more details on which can be found in [4,14,15]. In 1990, Hull and White [4] proposed a novel SDE such that the dynamics of the CIR process can be governed by time-depending parameters, which is called the ECIR process. The ECIR process is so attractive as a practical model to price the European bond option. In 2003, Egorov et al. [16] presented the transition PDF of the ECIR process

in a complicated form of the modified Bessel function of the first kind and proposed a method to receive a closed-form approximation of the transition PDF through the Hermite approximation. This is one of the most practically used empirical evidences to confirm that it is not easy to obtain the conditional expectation (2) by using the transition PDF of the ECIR process.

For this work, we assume that $x_s$ is governed by the ECIR process and $R(t, r)$ is a bounded continuous discount rate function; the value of the asset at initial time $t$ can be rewritten as $V_t = \mathbb{E}^Q \left[ e^{-\int_t^T R(s,x_s)\,ds} f(x_T) \mid \mathcal{F}_t \right]$, as can also be seen in [17]. With $R \equiv 0$ and $f(x) = x^\gamma$ where $\gamma \in \mathbb{R}$, under the probability measure $P$, Dufresne [18] derived a closed-form formula for the conditional moments, $\mathbb{E}^P \left[ x_T^\gamma \mid \mathcal{F}_t \right]$, for some sufficient conditions on $\gamma$ and the parameters in the CIR process $x_t$. In 2007, under a risk-neutral probability $Q$ based on the CIR process, Ben-Ameur et al. [1] estimated an ex-coupon holding value at time $T$, $\mathbb{E}^Q \left[ e^{-\int_t^T x_s\,ds} f(x_T) \mid \mathcal{F}_t \right]$, where $f$ denotes the value of the bond at time $T$. Their result is applicable but not in closed-form solutions. Moreover, they needed to find the joint distribution of the random vector $\left( x_{t+\delta}, \int_t^{t+\delta} x_t\,dt \right)$ where $\delta > 0$. This is characterized by its Laplace transform which can be described by the conditional expectation (2) when $\gamma = 0$ and $\beta = 0$. Recently, under the CIR process, Grasselli [2] directly determined a mathematical expression of the conditional expectation (2). However, their expression was expressed in terms of a product of the confluent hypergeometric function and the gamma functions. This may be hard to work with in some cases.

We move our focus onto the ECIR process. In 2016, under probability measures, an analytical formula was proposed by Rujivan [19] which was extended from Dufresne's approach [18] to the ECIR process for the case $\gamma \in \mathbb{R}$. In 2018, an explicit formula for the conditional expectations of a product of polynomial and exponential function, in the form $\mathbb{E}^P \left[ x_T^\gamma e^{-\lambda x_T} \mid \mathcal{F}_t \right]$, was analytically derived by Sutthimat et al. [20] for the case $\gamma, \lambda \in \mathbb{R}$. Their results cover the results in such formulas of the Rujivan's present [19] in the case of $\lambda = 0$. Indeed, both works on the ECIR process have a limitation. A major concern for their formulas in Theorems 1 and 2 of their works [19,20] is that the coefficients $A_{\gamma-k}(\tau)$ may not be integrable to receive the exact integrations. Some numerical methods for integrations are required to manipulate those integral terms in this very reason. However, both results presented in [19,20] did not provide any methods to overcome this issue. Both results are not ready for practical applications. In our analysis, we also present a numerical method to deal with this challenge.

The useful applications of (2) under the CIR and ECIR processes need to be mentioned. To price interest rate swaps (IRSs), a financial contingent claim, which is a financial derivative whose payoff depends on the uncertain future real value of other underlying assets, is assumed together to follow the CIR process. The IRS is one of the common types of contingent claim derivatives as a modified version of swaps. Normally, the cash flows of IRSs on the payment dates are the same as the forward rate agreements (FRAs) which are the contact that the forward rates can be fixed by an investor. In brief, an IRS is a form of series of FRAs. We give some interesting works under the assumption that the discount rate is continuously compounded, which have been well studied in the literature and can apply our result of (2) to those works. In 2004, Mallier and Alobaidi [21] supposed that the risk-neutral interest rates follow the CIR process. By utilizing the Green's function approach, they provided analytical expressions of the swap values for two well-known types of IRSs, which are the arrears and vanilla swaps. Their analytical expressions, a sum of values of the FRAs, which was in a closed form for an arrears swap but very complicated because it depends on the gamma and the Kummer's functions. However, for a vanilla swap, their result was not in closed form and much more complicated than the results of those arrears swaps. Unlike the results of Mallier and Alobaidi, Moreno and Platania [22] provided a mathematical formula for the FRA values in 2015 for a special case of the ECIR process, namely the cyclical square-root model; more details on this can be found in their Proposition 8. Thus, an interest rate swap valuation was straightforwardly obtained as a

consequence of this proposition. In fact, Proposition 8 in their work consists of the Mathieu cosine and sine functions [23] and the parameter $A(\tau)$ given in this proposition may not be exactly integrable. Thamrongrat and Rujivan [24] recently published an analytical formula for pricing IRSs in terms of bond prices based on the ECIR process which was performed under a discrete discount rate.

This paper successfully worked out an analytical formula of the conditional expectations (2) for the ECIR process in terms of analytical expression. Furthermore, their consequences were investigated without requiring the transition PDF of the ECIR process. Additionally, for the CIR process, the formulas were reduced to concise forms which give a greater advantage than the other approaches in the literature. Furthermore, the ECIR process-facilitated valuation of financial derivatives is provided by using our proposed results. Under ECIR process, this paper further suggests a numerical algorithm of the conditional expectations (2) in case the Riccati differential equation may not be exactly solved.

This paper is organized as follows. A brief overview of the CIR process as well as the ECIR process are provided in Section 2. The key methodology is mentioned in Section 3 to address the main relevant concept for our main result, which is an analytical formula of the conditional expectations (2) of the ECIR process. Section 4 gives a numerical method to work with the generalized Riccati differential equation and one major concerned limitation of our formula is discussed here. Section 5 validates our formulas and discusses the analytical formulas' advantages compared with the Monte Carlo (MC) simulations. In Section 6, some financial applications are demonstrated based on our proposed formula. The aim of this study is recapitulated and concluded in Section 7.

## 2. The Extended Cox–Ingersoll–Ross Process

In this paper, we assume that the interest rate $x_t$ follows the ECIR process under a risk-neutral probability measure $Q$, which is a diffusion model whose solution satisfies the following SDE [4],

$$dx_t = \theta(t)(\mu(t) - x_t)\, dt + \sigma(t)\sqrt{x_t}\, dW_t, \quad 0 \le t \le T. \tag{3}$$

The well-known $W_t$ is a Wiener process or Brownian motion whose increments are generated by the symmetry of mean zero Gaussian distribution. Sometimes, the parameters in (3) are referred to as follows: $\theta$ is the speed of adjustment to the long-term mean $\mu$, while $\sigma$ indicates to the state space of the diffusion. The two assumptions explored by Maghsoodi [15] are required to demonstrate that there is a path-wise unique strong solution for the ECIR process $x_t$ and to avoid zero a.e. with regard to the probability measure $P$ for a specified time $t$ during a time period $[0, T]$; more details on this can be found in Theorems 2.1 and 2.4 of [15]. We thus require the following sufficient condition.

**Assumption 1.** *Time parameters $\theta(t), \mu(t)$ and $\sigma(t)$ in (3) are smooth and strictly positive. The time function $\frac{\mu(t)}{\sigma^2(t)}$ is locally bounded and $2\theta(t)\mu(t) \ge \sigma(t)^2$ on $[0, T]$.*

To achieve our aim, a common question arises: why not directly use the transition PDF of the CIR process? It is known that its transition PDF has an expression in a form of Gamma density function and Laguerre polynomials; more details on this can be found in [25,26]. The transition PDF can be written in an explicit form as

$$p(x, T \mid x_t, t) = c_\tau\, e^{-(u+v)} \left(\frac{v}{u}\right)^{q/2} I_q\left(2\sqrt{uv}\right),$$

where $\tau = T - t, c_\tau = \frac{2\theta}{\sigma^2(1 - e^{-\theta\tau})}, u = c_\tau x_t e^{-\theta\tau}, vs. = c_\tau x, q = \frac{2\theta\mu}{\sigma^2} - 1$ and $I_q(\cdot)$ is the ordered $q$ Bessel function of the first kind,

$$I_q(x) = \sum_{k=0}^{\infty} \left(\frac{x}{2}\right)^{2k+q} \frac{1}{\Gamma(k+1)\Gamma(k+q+1)}.$$

Since the transition PDF is complicated, as shown above, solving the closed-form formulas for (2) by applying the transition PDF is more complicated.

It becomes even more difficult in the ECIR process, for example, as in the ECIR($d$) process observed by Egorov et al. in 2003 [16]. Its dynamics are followed by a time-inhomogeneous diffusion process as

$$dx_t = \theta \left( \frac{\sigma_0^2 d}{4\theta} e^{2\sigma_1 t} - x_t \right) dt + \sigma_0 e^{\sigma_1 t} \sqrt{x_t} \, dW_t,$$

where $\theta, \sigma_0$ are positive, $\sigma_1$ is real and $d$ is positive. Its transition PDF was first proposed by Maghsoodi [15],

$$p(x, T \mid x_t, t) = \frac{1}{2} G e^{-\frac{\lambda + Gr}{2}} \left( \frac{Gr}{\lambda} \right)^{\frac{d-2}{4}} I_{\frac{d}{2}-1}(\lambda Gr)$$

with $\lambda = x_t v$, $G = e^{\theta \tau} v$, $v = \frac{8\sigma_1}{\sigma_0^2} e^{-\theta \tau} \left( e^{2\sigma_1 T} - e^{2\sigma_1 t} \right)$, $\tau = T - t$ and again $I_q(\cdot)$ is the Bessel function of the first kind. To avoid using those of the transition PDFs for solving (2), this paper applies Feynman–Kac representation which offers a method for solving a conditional expectation of an Itô random process by deterministic implementations, more details on which can be found in [27–30].

## 3. Main Results

### 3.1. Closed-Form Formula: Conditional Expectation

The first contribution of this section is to provide an integral form formula for the pricing of the contingent claim with the polynomial payoff (2) by solving the PDE according to the Feynman–Kac representation.

**Theorem 1.** *Let $0 \leq t \leq T$ and $x_t$ follow the ECIR process (3) with $\alpha, \beta, \gamma \in \mathbb{R}$. Then,*

$$\mathbb{E}^Q \left[ x_T^\gamma e^{-\int_t^T (\alpha x_s + \beta) \, ds} \mid x_t = x \right] = e^{B(\tau)x} \sum_{j=0}^{\infty} A_j^{\langle \gamma \rangle}(\tau) x^{\gamma - j} =: U_E^\gamma(x, \tau), \qquad (4)$$

*for all $(x, \tau) \in D_E \subset (0, \infty) \times [0, \infty)$, where $\tau = T - t \geq 0$. Under the assumption that the infinite series in (4) uniformly converges on $D_E$, the coefficients in (4) can be expressed as*

$$\begin{aligned} A_0^{\langle \gamma \rangle}(\tau) &= e^{\int_0^\tau P_0(u) \, du}, \\ A_j^{\langle \gamma \rangle}(\tau) &= e^{\int_0^\tau P_j(u) \, du} \int_0^\tau e^{-\int_0^u P_j(s) \, ds} Q_j(u) A_{j-1}^{\langle \gamma \rangle}(u) \, du, \end{aligned} \qquad (5)$$

*for $j \in \mathbb{N}$, where*

$$\begin{aligned} P_j(\tau) &= \theta(T - \tau)\mu(T - \tau)B(\tau) - \theta(T - \tau)(\gamma - j) + \sigma^2(T - \tau)(\gamma - j)B(\tau) - \beta, \\ Q_j(\tau) &= (\gamma - j + 1)\left( \theta(T - \tau)\mu(T - \tau) + \frac{1}{2}\sigma^2(T - \tau)(\gamma - j) \right), \end{aligned} \qquad (6)$$

*and $B(\tau)$ can be obtained by solving the following Riccati differential equation*

$$B'(\tau) = \frac{1}{2}\sigma^2(T - \tau)B^2(\tau) - \theta(T - \tau)B(\tau) - \alpha, \quad B(0) = 0. \qquad (7)$$

**Proof.** The Feynman–Kac representation is used to solve $U := U_E^\gamma(x, \tau)$, which is defined as a series in (4) that satisfies the appropriate PDE under the uniformly convergent assumption. Thus, we have

$$-U_\tau + \theta(T - \tau)(\mu(T - \tau) - x)U_x + \frac{1}{2}x\sigma^2(T - \tau)U_{xx} - (\alpha x + \beta)U = 0 \qquad (8)$$

with the initial condition at $\tau = 0$

$$U_E^\gamma(x,0) = \mathbb{E}^Q\left[x_T^\gamma e^{-\int_T^T(\alpha x_s + \beta)\,ds} \mid x_T = x\right] = x^\gamma. \tag{9}$$

Comparing the coefficients at $\tau = 0$ in (4) and (9) to receive the initial conditions $B(0) = 0$, $A_0^{\langle\gamma\rangle}(0) = 1$ and $A_j^{\langle\gamma\rangle}(0) = 0$ for $j \in \mathbb{N}$. Then, we compute (8) using (4) to find the partial derivatives $U_\tau$, $U_x$ and $U_{xx}$. Consequently, we have

$$
\begin{aligned}
0 = &- e^{B(\tau)x}\left(\sum_{j=0}^\infty A_j^{\langle\gamma\rangle\prime}(\tau)x^{\gamma-j} + xB'(\tau)\sum_{j=0}^\infty A_j^{\langle\gamma\rangle}(\tau)x^{\gamma-j}\right) \\
&+ \theta(T-\tau)(\mu(T-\tau) - x)e^{B(\tau)x}\left(\sum_{j=0}^\infty(n-j)A_j^{\langle\gamma\rangle}(\tau)x^{\gamma-j-1} + B(\tau)\sum_{j=0}^\infty A_j^{\langle\gamma\rangle}(\tau)x^{\gamma-j}\right) \\
&+ \frac{1}{2}x\sigma^2(T-\tau)e^{B(\tau)x}\left(\sum_{j=0}^\infty(n-j)(n-j-1)A_j^{\langle\gamma\rangle}(\tau)x^{\gamma-j-2} + 2B(\tau)\sum_{j=0}^\infty(n-j)A_j^{\langle\gamma\rangle}(\tau)x^{\gamma-j-1}\right. \\
&\left. + B^2(\tau)\sum_{j=0}^\infty A_j^{\langle\gamma\rangle}(\tau)x^{\gamma-j}\right) - (\alpha x + \beta)e^{B(\tau)x}\sum_{j=0}^\infty A_j^{\langle\gamma\rangle}(\tau)x^{\gamma-j},
\end{aligned}
\tag{10}
$$

which can be simplified as

$$
\begin{aligned}
0 = &- \left(A_0^{\langle\gamma\rangle}(\tau)\left(B'(\tau) + \theta(T-\tau)B(\tau) - \frac{1}{2}\sigma^2(T-\tau)B^2(\tau) + \alpha\right)\right)x^{\gamma+1} \\
&+ \left(-A_0^{\langle\gamma\rangle\prime}(\tau) + P_0(\tau)A_0^{\langle\gamma\rangle}(\tau) - A_1^{\langle\gamma\rangle}(\tau)\left(B'(\tau) + \theta(T-\tau)B(\tau) - \frac{1}{2}\sigma^2(T-\tau)B^2(\tau) + \alpha\right)\right)x^\gamma \\
&+ \sum_{j=1}^\infty\left(-A_j^{\langle\gamma\rangle\prime}(\tau) + P_j(\tau)A_j^{\langle\gamma\rangle}(\tau) + Q_j(\tau)A_{j-1}^{\langle\gamma\rangle}(\tau)\right. \\
&\left. - A_{j+1}^{\langle\gamma\rangle}(\tau)\left(B'(\tau) + \theta(T-\tau)B(\tau) - \frac{1}{2}\sigma^2(T-\tau)B^2(\tau) + \alpha\right)\right)x^{\gamma-j}
\end{aligned}
\tag{11}
$$

Considering (11) as a series solution in $x$, we receive the system of ODEs as follows,

$$
\begin{aligned}
0 &= A_0^{\langle\gamma\rangle\prime}(\tau) - P_0(\tau)A_0^{\langle\gamma\rangle}(\tau), \\
0 &= A_j^{\langle\gamma\rangle\prime}(\tau) - P_j(\tau)A_j^{\langle\gamma\rangle}(\tau) - Q_j(\tau)A_{j-1}^{\langle\gamma\rangle}(\tau), \\
0 &= B'(\tau) + \theta(T-\tau)B(\tau) - \frac{1}{2}\sigma^2(T-\tau)B^2(\tau) + \alpha,
\end{aligned}
$$

with their initial conditions $B(0) = 0$, $A_0^{\langle\gamma\rangle}(0) = 1$ and $A_j^{\langle\gamma\rangle}(0) = 0$ for $j \in \mathbb{N}$. Thus, the solutions of the ODEs are given in (5) and (7) as required. $\square$

Notice that the function $B$ is the solution of a generalized Riccati differential equation. It is well known that its solution is not available and can be only treated in some cases. To overcome this problem, one can employ the numerical method discussed in the following section.

By examining (4) when $\gamma = n \in \mathbb{N}_0$ and $\gamma = m - \frac{2\theta(\tau)\mu(\tau)}{\sigma^2(\tau)}$ for $m \in \mathbb{N}$, the infinite sum in (4) is terminated at order $n$ and can be written as in the following theorems.

**Corollary 1.** *According to Theorem 1 with $\gamma = n \in \mathbb{N}_0$, we have*

$$U_E^n(x,\tau) = \mathbb{E}^Q\left[x_T^n e^{-\int_t^T(\alpha x_s + \beta)\,ds} \mid x_t = x\right] = e^{B(\tau)x}\sum_{j=0}^n A_j^{\langle n\rangle}(\tau)x^{n-j}, \tag{12}$$

*for all $(x,\tau) \in D_E \subset (0,\infty) \times [0,\infty)$, $\tau = T - t \geq 0$, where the coefficients $A_j^{\langle n\rangle}(\tau)$ are defined by (5) and (6), and $B(\tau)$ is the solution of (7).*

**Proof.** Considering (5) when $j = n + 1$ obtains $Q_j(\tau) = Q_{n+1}(\tau) = 0$, it implies that the coefficient $A_j^{\langle n \rangle}(\tau) = A_{n+1}^{\langle n \rangle}(\tau) = 0$. Since the coefficient $A_j^{\langle n \rangle}(\tau)$ is a type of recurrence problem involving the initial condition $A_{n+1}^{\langle n \rangle}(\tau) = 0$, $A_j^{\langle n \rangle}(\tau) = 0$ for all $j \geq n + 1$. Thus, the infinite sum (4) can be reduced to the finite sum as shown in (12). □

**Corollary 2.** *According to Theorem 1 with $\gamma = m - \frac{2\theta(\tau)\mu(\tau)}{\sigma^2(\tau)}$ for all $\tau \geq 0$, $m \in \mathbb{N}$, we have*

$$U_E^\gamma(x, \tau) = \mathbb{E}^Q\left[x_T^\gamma e^{-\int_t^T (\alpha x_s + \beta)\, ds} \mid x_t = x\right] = e^{B(\tau)x} \sum_{j=0}^m A_j^{\langle \gamma \rangle}(\tau) x^{\gamma - j}, \tag{13}$$

*for $(x, \tau) \in D_E \subset (0, \infty) \times [0, \infty)$, $\tau = T - t \geq 0$, where the coefficients $A_j^{\langle \gamma \rangle}(\tau)$ are defined by (5) and (6), and $B(\tau)$ is the solution of (7).*

**Proof.** The proof can be shown similarly as in Corollary (1) by considering (5) and (6). □

It is worth mentioning that our proposed formulas for the ECIR process generalize some results in the literature. Furthermore, this section provides concise formulas for the CIR process reduced from the previous theorems where the parameters depending on time $\theta(t) = \theta$, $\mu(t) = \mu$ and $\sigma(\tau) = \sigma$ are constant functions. In this case, Riccati differential Equation (7) is solvable; thus, integral functions can be exactly solved as shown in the following theorems.

**Corollary 3.** *Suppose that $x_t$ follows the CIR process with $\alpha \geq -\frac{\theta^2}{2\sigma^2}$, $\beta, \gamma \in \mathbb{R}$. Let $0 \leq t \leq T$. Then,*

$$U_C^\gamma(x, \tau) := \mathbb{E}^Q\left[x_T^\gamma e^{-\int_t^T (\alpha x_s + \beta)\, ds} \mid x_t = x\right] = e^{B(\tau)x} \sum_{j=0}^\infty A_j^{\langle \gamma \rangle}(\tau) x^{\gamma - j}, \tag{14}$$

*for all $(x, \tau) \in D_C \subset (0, \infty) \times [0, \infty)$, $\tau = T - t \geq 0$. Under the assumption that the infinite series in (14) uniformly converges on $D_C$, the coefficients in (14) can be expressed as*

$$A_0^{\langle \gamma \rangle}(\tau) = H_0(\tau),$$
$$A_j^{\langle \gamma \rangle}(\tau) = H_j(\tau)\left(\prod_{k=1}^j \frac{2Q_k}{k}\right)\left(\frac{e^{\rho\tau} - 1}{(\rho - \theta) + e^{\rho\tau}(\rho + \theta)}\right)^j, \tag{15}$$

*for $j \in \mathbb{N}$, where $\rho = \sqrt{\theta^2 + 2\alpha\sigma^2}$ and*

$$Q_j = (\gamma - j + 1)\left(\theta\mu + \frac{1}{2}\sigma^2(\gamma - j)\right),$$
$$H_j(\tau) = \exp\left[\left(\frac{\theta^2\mu}{\sigma^2} - \beta + \left((\gamma - j) + \frac{\theta\mu}{\sigma^2}\right)\rho\right)\tau\right]\left(\frac{2\rho}{(\rho - \theta) + e^{\rho\tau}(\rho + \theta)}\right)^{2\left((\gamma - j) + \frac{\theta\mu}{\sigma^2}\right)}. \tag{16}$$

*In addition, function $B$ given in (7) can be solved as*

$$B(\tau) = -\frac{2\alpha(e^{\rho\tau} - 1)}{\rho(e^{\rho\tau} + 1) + \theta(e^{\rho\tau} - 1)}. \tag{17}$$

**Proof.** Determining (7) with constant parameters $\theta(t) = \theta$, $\mu(t) = \mu$ and $\sigma(\tau) = \sigma$, the explicit solution for the Riccati differential Equation (7) is

$$B(\tau) = -\frac{2\alpha(e^{\rho\tau} - 1)}{\rho(e^{\rho\tau} + 1) + \theta(e^{\rho\tau} - 1)}, \tag{18}$$

where $\rho = \sqrt{\theta^2 + 2\alpha\sigma^2}$, and its integration is

$$\int_0^\tau B(u)\, du = \frac{2}{\sigma^2} \ln \frac{2\rho e^{(\frac{\rho+\theta}{2})\tau}}{(\rho - \theta) + e^{\rho\tau}(\rho + \theta)}.$$

Thus, we have

$$e^{\int_0^\tau P_j(u)\, du} = \exp\left[\int_0^\tau \left(\theta\mu B(u) - \theta(\gamma - j) + \sigma^2(\gamma - j)B(u) - \beta\right)du\right]$$

$$= \exp\left[\left(\frac{\theta^2\mu}{\sigma^2} - \beta + \left((\gamma - j) + \frac{\theta\mu}{\sigma^2}\right)\rho\right)\tau\right]\left(\frac{2\rho}{(\rho - \theta) + e^{\rho\tau}(\rho + \theta)}\right)^{2\left((\gamma - j) + \frac{\theta\mu}{\sigma^2}\right)}, \quad (19)$$

which are defined as $H_j(\tau)$ for $j \in \mathbb{N}_0$. From Theorem 1, $A_0^{\langle\gamma\rangle}(\tau) = e^{\int_0^\tau P_0(u)\, du} = H_0(\tau)$ and

$$A_1^{\langle\gamma\rangle}(\tau) = H_1(\tau)\int_0^\tau \frac{1}{H_1(u)} Q_1 A_0^{\langle\gamma\rangle}(u)\, du = 2H_1(\tau)Q_1\left(\frac{e^{\rho\tau} - 1}{(\rho - \theta) + e^{\rho\tau}(\rho + \theta)}\right).$$

From the result presented in (5) for $j \in \mathbb{N}$, we obtain

$$A_j^{\langle\gamma\rangle}(\tau) = H_j(\tau)\int_0^\tau \frac{1}{H_j(u)} Q_j A_{j-1}^{\langle\gamma\rangle}(u)\, du$$

$$= H_j(\tau)\int_0^\tau \frac{1}{H_j(u)} Q_j \left(H_{j-1}(u)\left(\prod_{k=1}^{j-1} \frac{2Q_k}{k}\right)\left(\frac{e^{\rho u} - 1}{(\rho - \theta) + e^{\rho u}(\rho + \theta)}\right)^{j-1}\right)du$$

$$= H_j(\tau)\left(\prod_{k=1}^{j-1} \frac{2Q_k}{k}\right)\int_0^\tau e^{\rho u}\left(\frac{2\rho}{(\rho - \theta) + e^{\rho u}(\rho + \theta)}\right)^2\left(\frac{e^{\rho u} - 1}{(\rho - \theta) + e^{\rho u}(\rho + \theta)}\right)^{j-1}du$$

$$= H_j(\tau)\left(\prod_{k=1}^{j} \frac{2Q_k}{k}\right)\left(\frac{e^{\rho\tau} - 1}{(\rho - \theta) + e^{\rho\tau}(\rho + \theta)}\right)^j.$$

Under the uniformly convergent assumption, this completes the proof. $\square$

The case in which $\gamma = n \in \mathbb{N}_0$, $U_C^\gamma(x, \tau)$ in (14) can be expressed as a finite term of a power series in $x$ as follows.

**Corollary 4.** *According to Corollary 3 with $\gamma = n \in \mathbb{N}_0$, we have*

$$U_C^n(x, \tau) = \mathbb{E}^Q\left[x_T^n e^{-\int_t^T(\alpha x_s + \beta)\, ds} \mid x_t = x\right] = e^{B(\tau)x} \sum_{j=0}^n A_j^{\langle n \rangle}(\tau)x^{n-j}, \quad (20)$$

*for $(x, \tau) \in D_C \subset (0, \infty) \times [0, \infty)$, $\tau = T - t \geq 0$, where the coefficients $A_j^{\langle n \rangle}(\tau)$ and $B(\tau)$ are defined by (15), (16) and (17).*

**Proof.** The proof is rather trivial by combining Corollary 1 with Corollary 3. $\square$

Furthermore, in this case and in that where $\gamma = m - \frac{2\theta\mu}{\sigma^2}$ for $m \in \mathbb{N}$, the following theorem can readily be reduced from Corollary 3.

**Corollary 5.** *According to Corollary 3 with $\gamma = m - \frac{2\theta\mu}{\sigma^2}$ for $m \in \mathbb{N}$, we have*

$$U_C^\gamma(x, \tau) = \mathbb{E}^Q\left[x_T^\gamma e^{-\int_t^T(\alpha x_s + \beta)\, ds} \mid x_t = x\right] = e^{B(\tau)x} \sum_{j=0}^m A_j^{\langle\gamma\rangle}(\tau)x^{\gamma-j}, \quad (21)$$

*for $(x, \tau) \in D_C \subset (0, \infty) \times [0, \infty)$, $\tau = T - t \geq 0$, where coefficients $A_j^{\langle\gamma\rangle}(\tau)$ and $B(\tau)$ are defined by (15), (16) and (17).*

**Proof.** The proof is relatively trivial by combining Corollary 2 with Corollary 3. □

**Remark 1.** *Consequently, our approach formulas can be extended to calculate the mixed polynomial payoff. By applying the tower property for $0 \le t < s \le T$ where $\tau_1 = s - t$ and $\tau_2 = T - s$, the price of the T-claim with mixed polynomial payoff for ECIR process* (3) *can be expressed as*

$$\mathbb{E}^Q\left[ x_s \, x_T \, e^{-\int_t^T (\alpha x_u + \beta)\, du} \mid x_t = x \right] = \mathbb{E}^Q\left[ x_s \, e^{-\int_t^s (\alpha x_u + \beta)\, du} \, \mathbb{E}^Q\left[ x_T \, e^{-\int_s^T (\alpha x_u + \beta)\, du} \mid x_s \right] \mid x_t = x \right].$$

**Remark 2.** *The benefits of this work, when $\alpha = \beta = 0$, can be performed by special cases of our proposed theorems, such as the first and the second conditional moments, variance, central moment, mixed moment, covariance, and correlation.*

*3.2. Closed-Form Formula: Unconditional Expectation*

Under Assumption 1, this section proposes two corollaries which are deduced from the conditional formula for the CIR process presented in Corollary 4 to the unconditional formula whereas $\tau \to \infty$. It should be noted that the following formulas are no longer dependent on the given value of $x$.

**Corollary 6.** *Suppose that $x_t$ follows the CIR process and $\alpha, \beta > 0$. The price of the T-claim with polynomial payoff at equilibrium, for all $n \in \mathbb{N}_0$, $x > 0$ and $\tau = T - t \ge 0$, is*

$$\lim_{\tau \to \infty} U_C^n(x, \tau) = \lim_{T \to \infty} \mathbb{E}^Q\left[ x_T^n e^{-\int_t^T (\alpha x_s + \beta)\, ds} \mid x_t = x \right] = 0.$$

**Proof.** By considering $H_j(\tau)$ in (16), for all $j = 0, 1, 2, \ldots, n$, it can be rewritten as

$$H_j(\tau) = \left( \frac{2\rho \exp\left[ \left( \frac{\rho}{2} + \frac{\frac{\theta^2 \mu}{\sigma^2} - \beta}{2\left((n-j) + \frac{\theta\mu}{\sigma^2}\right)} \right) \tau \right]}{(\rho - \theta) + e^{\rho\tau}(\rho + \theta)} \right)^{2\left((n-j) + \frac{\theta\mu}{\sigma^2}\right)}.$$

It is not difficult to see that $\left( \frac{\rho}{2} + \frac{\frac{\theta^2 \mu}{\sigma^2} - \beta}{2\left((n-j) + \frac{\theta\mu}{\sigma^2}\right)} \right) < \rho$ for all $j$. So, $\lim\limits_{\tau \to \infty} H_j(\tau) = 0$ and then

$$\lim_{\tau \to \infty} A_j(\tau) = \left( \prod_{k=1}^{j} \frac{2Q_k}{k} \right) \lim_{\tau \to \infty} \left( H_j(\tau) \left( \frac{e^{\rho\tau} - 1}{(\rho - \theta) + e^{\rho\tau}(\rho + \theta)} \right)^j \right) = 0,$$

for all $j = 0, 1, 2, \ldots, n$. Thus, $\lim\limits_{\tau \to \infty} U_C^n(x, \tau) = 0$. □

**Corollary 7.** *Suppose that $x_t$ follows the CIR process and $\alpha, \beta = 0$. The price of the T-claim with polynomial payoff at equilibrium, for all $n \in \mathbb{N}_0$, $x > 0$ and $\tau = T - t \ge 0$, is*

$$\mathbb{U}_n^\infty := \lim_{\tau \to \infty} U_C^n(x, \tau) = \lim_{T \to \infty} \mathbb{E}^Q[x_T^n \mid x_t = x] = \prod_{k=1}^{n} \frac{2\theta\mu + \sigma^2(n-k)}{2\theta}. \tag{22}$$

**Proof.** According to the (20) in Corollary 4 with $\alpha, \beta = 0$, it is then obtained $\rho = \theta$. As $\tau \to \infty$, we note that $\lim_{\tau \to \infty} H_n(\tau) = 1$ and $\lim_{\tau \to \infty} H_j(\tau) = 0$ for all $j = 0, 1, \ldots, n-1$. In other words, the coefficient terms $A_j(\tau)$ of $x^{n-j}$ approach 0, for $k = 0, 1, \ldots, n-1$. The

remaining term, $j = n$, is only determined. From (17) in Corollary 3, since $\alpha = 0$, $B(\tau) = 0$ and yields

$$\mathbb{U}_n^\infty = \lim_{\tau \to \infty} A_n(\tau) = \lim_{\tau \to \infty} H_n(\tau) \left( \prod_{k=1}^n \frac{2Q_k}{k} \right) \left( \frac{e^{\theta\tau} - 1}{2\theta e^{\theta\tau}} \right)^n = \prod_{k=1}^n \frac{2\theta\mu + \sigma^2(n-k)}{2\theta},$$

as required.  $\square$

**Remark 3.** *After some algebraic manipulations, we can show that Rujivan's formula [19] reduces to our formula* (22).

**Remark 4.** *It is obvious to what extent Corollary 4 applies to unconditional cases when $\alpha, \beta \geq 0$ as displayed in Corollaries 6 and 7. In particular, it is easily checked that, when $\beta < 0$, $\lim_{\tau \to \infty} U_C^n(x, \tau)$ converges towards 0 if and only if $\frac{\theta^2\mu - \sigma^2\beta}{\theta\mu} < \rho$.*

### 3.3. Analysis of Convergence

The aim of this section is to investigate Corollary 3 and whether the infinite sum given in (14) converges. Notice that series (14) converges whenever $A_j^{\langle\gamma\rangle} = 0$ for some $j \in \mathbb{N}_0$. To see that the factor of $A_j^{\langle\gamma\rangle}$ in (15) only depends on $Q_j$ which can be zero. In other words, the series (14) converges if and only if $Q_j = 0$ for some $j \in \mathbb{N}_0$. There are only two cases that $Q_j = 0$, i.e., $\gamma = n \in \mathbb{N}_0$ and $\gamma = m - \frac{2\theta\mu}{\sigma^2}$ for $m \in \mathbb{N}$. The convergence cases of (14) are already provided in Corollaries 4 and 5. However, the case of $Q_j \neq 0$ for any $j \in \mathbb{N}_0$, and the series (14) diverges as follows.

$$\lim_{n \to \infty} \left| \frac{A_{n+1}^{\langle\gamma\rangle}(\tau) x^{\gamma-n-1}}{A_n^{\langle\gamma\rangle}(\tau) x^{\gamma-n}} \right| = \lim_{n \to \infty} \left| \frac{\frac{2^{n+1}}{(n+1)!} H_{n+1}(\tau) \left( \prod_{k=1}^{n+1} Q_k \right) \left( \frac{e^{\rho\tau} - 1}{(\rho-\theta) + e^{\rho\tau}(\rho+\theta)} \right)^{n+1} x^{\gamma-n-1}}{\frac{2^n}{n!} H_n(\tau) \left( \prod_{k=1}^n Q_k \right) \left( \frac{e^{\rho\tau} - 1}{(\rho-\theta) + e^{\rho\tau}(\rho+\theta)} \right)^n x^{\gamma-n}} \right|$$

$$= \lim_{n \to \infty} \left| \frac{2e^{-\rho\tau} Q_{n+1}}{x(n+1)} \left( \frac{e^{\rho\tau} - 1}{(\rho-\theta) + e^{\rho\tau}(\rho+\theta)} \right) \left( \frac{2\rho}{(\rho-\theta) + e^{\rho\tau}(\rho+\theta)} \right)^{-2} \right|.$$

Since $Q_{n+1}$ is a second-degree polynomial in $n$, the above expression is $\mathcal{O}(n)$, thus, by ratio test, the (14) diverges.

### 4. Numerical Procedures

The valuation of the contingent claim with polynomial payoff based on the ECIR process through Theorem 1, the formula (4), is an infinite sum of coefficients in (5). These coefficients are defined as in the integral forms and depend on many parameters. Under certain circumstances, i.e., when parameters are complicated, the integral cannot be precisely evaluated, or when the Riccati differential Equation (7) cannot be solved directly. Thus, numerical methods are required to approximate the coefficients. In this section, we numerically investigate the coefficients in (5) by utilizing numerical schemes based on the symmetry concept to approximate the formula (4).

Let us first consider the Riccati differential Equation (7). From Corollary 3, if the Riccati differential equation has constant coefficients, it has the exact solution, as shown in (18). However, if it has variable coefficients, the analytical solution is not easily obtained. In this case, one needs to approximate the solution by a numerical method; for example, in this work, we use the fourth-order Runge–Kutta (RK4) method [31]. Thus, we are concerned with the following initial value problem:

$$B'(s) = \frac{1}{2}\sigma^2(T-s)B^2(s) - \theta(T-s)B(s) - \alpha, \quad B(0) = 0 \tag{23}$$

for $s \in [0, \tau]$. We uniformly divide $[0, \tau]$ into $m$ subintervals generated by $s_i = ih$, $i = 0, 1, \ldots, m$, where $h = \frac{\tau}{m}$ is the step size. Then, we denote (23) by

$$f(s, B) := \frac{1}{2}\sigma^2(T - s)B^2 - \theta(T - s)B - \alpha.$$

Let $B_i = B(s_i)$; then $B_0 = 0$. By employing the RK4 method, we have four increments as follows:

$$\begin{aligned}
k_1 &= hf(s_i, B_i), \\
k_2 &= hf\left(s_i + \tfrac{h}{2}, B_i + \tfrac{k_1}{2}\right), \\
k_3 &= hf\left(s_i + \tfrac{h}{2}, B_i + \tfrac{k_2}{2}\right), \\
k_4 &= hf(s_i + h, B_i + k_3),
\end{aligned}$$

thus, we obtain that

$$B_{i+1} = B_i + \frac{1}{6}(k_1 + 2k_2 + 2k_3 + k_4).$$

Now, we have the approximate solutions $B_i$ of the Riccati differential equation at each nodal point $s_i \in [0, \tau]$, $i = 0, 1, \ldots, m$. We denote $\mathbf{B} = [B_0, B_1, \ldots, B_m]^\top$. Afterwards, this vector solution $\mathbf{B}$ is used to estimate the coefficients in (5). Since (5) is in integral form, in this work, we construct matrix representation for integration based on the concept of trapezoidal rule. By considering an integral function from the initial point $s_0$ to each point $s_i$, $i = 0, 1, \ldots, m$, it is approximated by the trapezoidal rule. We obtain:

$$F(s_0) := \int_{s_0}^{s_0} f(\xi)d\xi \approx 0,$$

$$F(s_1) := \int_{s_0}^{s_1} f(\xi)d\xi \approx \frac{h}{2}[f(s_0) + f(s_1)],$$

$$F(s_2) := \int_{s_0}^{s_2} f(\xi)d\xi \approx \frac{h}{2}[f(s_0) + 2f(s_1) + f(s_2)],$$

$$\vdots$$

$$F(s_m) := \int_{s_0}^{s_m} f(\xi)d\xi \approx \frac{h}{2}[f(s_0) + 2f(s_1) + \cdots + 2f(s_{m-1}) + f(s_m)].$$

From these integrations, we can construct the integration matrix by

$$\begin{bmatrix} F(s_0) \\ F(s_1) \\ F(s_2) \\ \vdots \\ F(s_m) \end{bmatrix} = \begin{bmatrix} 0 & & & & \\ \frac{h}{2} & \frac{h}{2} & & & \\ \frac{h}{2} & h & \frac{h}{2} & & \\ \vdots & \vdots & \ddots & \ddots & \\ \frac{h}{2} & h & \cdots & h & \frac{h}{2} \end{bmatrix} \begin{bmatrix} f(s_0) \\ f(s_1) \\ f(s_2) \\ \vdots \\ f(s_m) \end{bmatrix}$$

and denote this by $\mathbf{F} = \mathbf{J}\mathbf{f}$. This $\mathbf{J}$ is called the integration matrix, which is easily computed. We will then approximate the integral terms of $A_j^{\langle\gamma\rangle}(\tau)$ for $j \in \mathbb{N}_0 =: \mathbb{N} \cup \{0\}$ in (5) using the integration matrix $\mathbf{J}$, and we have

$$\mathbf{A}_0^{\langle\gamma\rangle} = e^{\mathbf{J}\mathbf{P}_0},$$

$$\mathbf{A}_j^{\langle\gamma\rangle} = e^{\mathbf{J}\mathbf{P}_j} \odot \left[\mathbf{J}\left(e^{-\mathbf{J}\mathbf{P}_j} \odot \mathbf{Q}_j \odot \mathbf{A}_{j-1}^{\langle\gamma\rangle}\right)\right],$$

where $\mathbf{A}_j^{\langle\gamma\rangle} = \left[A_j^{\langle\gamma\rangle}(s_0), A_j^{\langle\gamma\rangle}(s_1) \ldots, A_j^{\langle\gamma\rangle}(s_m)\right]^\top$, $\mathbf{P}_j = [P_j(s_0), P_j(s_1), \ldots, P_j(s_m)]^\top$ and $\mathbf{Q}_j = [Q_j(s_0), Q_j(s_1), \ldots, Q_j(s_m)]^\top$; the elements of $\mathbf{P}_j$ and $\mathbf{Q}_j$ can be directly calculated

by (6). The notation $\odot$ is the Hadamard product defined in [32] as the product of element-wise at the same positions in matrices. In this work, we use the exponential function of a matrix to denote the matrix whose element is the exponential of the element in that component.

Finally, we obtain the numerical formula for the pricing of the *T*-claim with the polynomial payoff (4) by

$$U_E^{\langle\gamma\rangle}(x,\tau) \approx e^{B(s_m)x} \sum_{j=0}^{\infty} A_j^{\langle\gamma\rangle}(s_m) x^{\gamma-j},$$

where $B(s_m)$ and $A_j^{\langle\gamma\rangle}(s_m)$ are the last components of vector solutions **B** and $\mathbf{A}_j^{\langle\gamma\rangle}$ described above, respectively. Moreover, we can reduce the number of computational points *m*, but still preserve the accuracy by using other numerical integration approaches such as Simpson's rule, Newton–Cotes, quadrature formula, etc., as can be seen in [33] for more details and references.

## 5. Experimental Validations

In this section, we verify the formula proposed in Section 3 by comparing with the Monte Carlo (MC) simulation based on the following ECIR process,

$$dx_t = \theta\left(\frac{\sigma_0^2 d e^{2\sigma_1 t}}{4\theta} - x_t\right)dt + \sigma_0 e^{\sigma_1 t}\sqrt{x_t}\,dW_t. \tag{24}$$

Comparing with (3), we use $\theta(t) = \theta$, $\mu(t) = \frac{\sigma_0^2 d e^{2\sigma_1 t}}{4\theta}$ and $\sigma(t) = \sigma_0 e^{\sigma_1 t}$, where $\theta, \sigma_0$ are positive numbers, $\sigma_1$ is a real number and with an integer $d \geq 2$; thus, the Assumption 1 is satisfied. In general, the expectation for the ECIR process (24) may be directly computed using the transition density of $x_t$ presented by Egorov et al. [16]. However, this approach will not produce an accurate value of $U_E^{\langle\gamma\rangle}(x,\tau)$ for a small $\tau$, and to overcome this problem, the MC simulation is employed to approximate the value of $U_E^{\langle\gamma\rangle}(x,\tau)$.

The MC simulations presented in this paper are based on the EM scheme which are implemented by the MATLAB software to receive numerical solutions of (24) for evaluating (2). In this case, we use the trapezoidal integration for the integral term. MATLAB R2020a and a laptop with the following specifications were used in all of our calculations: Windows 10 Education, 64-bit Operating System, Intel(R) Core(TM) i7-8550U, CPU @1.80GHz, 8.0 GB RAM.

### 5.1. Closed-Form Formulas for CIR and ECIR Processes with MC Simulations

In this experiment, Euler–Maruyama (EM) discretization was applied for the ECIR process (24). Higham and Mao proved the accuracy of approximations by the EM scheme [34]. According to (24), we used $\theta = 1$, $d = 2$, $\sigma_0 = 1$, and in particular, we set $\sigma_1 = 0, 1$ for the CIR and ECIR processes, respectively.

**Example 1** (CIR case, $\sigma_1 = 0$). *Formula* (20) *with* $\alpha = \beta = 0.01$ *and for* $\gamma = n = 1, 2$:

This example demonstrates the closed-form Formula (20) based on (24) in the case of $\gamma = n \in \mathbb{N}_0$ and $\sigma_1 = 0$, CIR process. To validate Formula (20) in Corollary 4, we use parameters $\theta = 1$, $\sigma_0 = 0.1$ and $d = 2$ in the process (24), and employ MC simulations with different initial values $x = 0.1, 0.2, \ldots, 1$ to generate $10,000$ sample paths of $x_t$, where each path consists of $10,000$ time steps over two different time intervals $[0, 0.1]$ and $[0, 1]$. The validations are performed through the comparisons between the Formula (20) and MC simulations based on $\gamma = 1, 2$.

As presented in Figure 1, the numerical results from MC simulations (colored circles) match completely with the results from Formula (20) (solid lines) for each $x = 0.1, 0.2, \ldots, 1$. Thus, the agreement of the results has validated the accuracy of Formula (20).

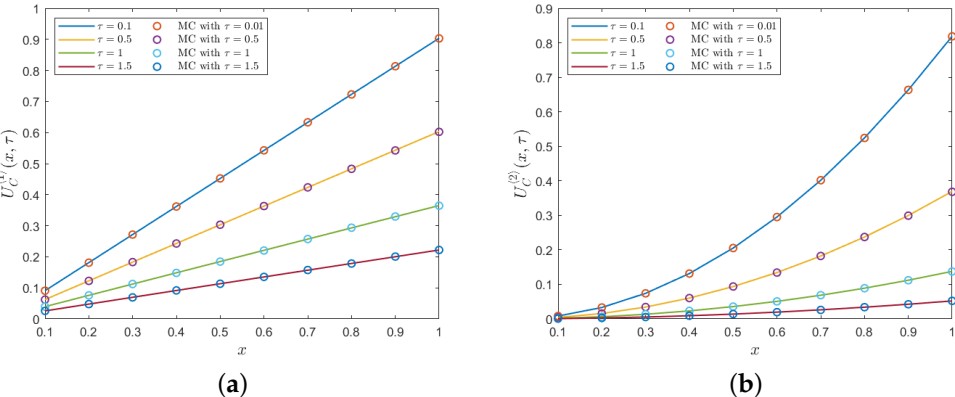

**Figure 1.** The validations of $U_C^n(x, \tau)$ for $\tau = 0.1, 0.5, 1, 1.5$ and $x = 0.1, 0.2, \ldots, 1.0$: (**a**) the first conditional moments; and (**b**) the second conditional moments.

**Example 2** (ECIR case, $\sigma_1 = 1$). *Formula* (12) *with* $\alpha = \beta = 1$ *and for* $\gamma = n = 1, 2$:

In this example, we validate the formula (12) of $U_E^n(x, \tau)$ based on the process (24) in the case of $\gamma = n \in \mathbb{N}_0$ with parameters $\theta = 1$, $\sigma_0 = \sigma_1 = 1$ and $d = 2$ by comparing with the MC simulation. Since the parameters are no longer constant, the solution $B(\tau)$ of (7) cannot be solved analytically; thus, the numerical scheme RK4 (see Section 4) is applied to approximate the solution $B(\tau)$ with 500 step sizes and the integral terms of the coefficients in (5) are approximated by the trapezoidal rule using 500 subintervals. MC simulations are performed using a number of sample paths, including 5000, 10,000, 20,000 and 40,0000, where each path consists of 10,000 steps over different time intervals $[0, \tau]$ for $\tau = 0.01, 0.1, 1, 2$.

Table 1 shows the comparisons between the numerical formula (12) of $U_E^n(x, \tau)$ and MC simulation for $\gamma = 1, 2$ at each initial value $x = 0.1, 0.2, \ldots, 1.0$. The accuracy is measured by the mean absolute differences (MADs) over initial values $x$. Table 1 shows that the obtained MADs are very small and become smaller as the number of sample paths increases for all cases of $\gamma$ and $T$. This result confirms the accuracy of the formula (12) as compared with the MC simulations. The average run times (ARTs) of the MC simulations for different numbers of paths are displayed in Table 1. Obviously, the ARTs of MC simulations are much more than that from the formula, which is approximately 0.3 s, especially when using a large number of paths.

**Table 1.** The MADs between estimated solutions of Formula (12) and MC simulations.

| $\gamma$ | No. of Paths | $\tau$ | | | | ART (s) |
|---|---|---|---|---|---|---|
| | | 0.01 | 0.1 | 1 | 2 | |
| 1 | 5000 | $8.756 \times 10^{-4}$ | $2.428 \times 10^{-3}$ | $1.884 \times 10^{-3}$ | $4.578 \times 10^{-4}$ | 18.26 |
| | 10,000 | $5.149 \times 10^{-4}$ | $1.386 \times 10^{-3}$ | $1.013 \times 10^{-3}$ | $3.746 \times 10^{-4}$ | 39.23 |
| | 20,000 | $3.129 \times 10^{-4}$ | $8.440 \times 10^{-4}$ | $7.492 \times 10^{-3}$ | $2.363 \times 10^{-4}$ | 73.19 |
| | 40,000 | $1.791 \times 10^{-4}$ | $7.463 \times 10^{-4}$ | $6.305 \times 10^{-4}$ | $1.436 \times 10^{-4}$ | 134.27 |
| 2 | 5000 | $1.599 \times 10^{-3}$ | $3.162 \times 10^{-3}$ | $9.097 \times 10^{-3}$ | $6.633 \times 10^{-3}$ | 18.34 |
| | 10,000 | $6.147 \times 10^{-4}$ | $2.182 \times 10^{-3}$ | $5.019 \times 10^{-3}$ | $3.873 \times 10^{-3}$ | 39.35 |
| | 20,000 | $2.980 \times 10^{-4}$ | $1.643 \times 10^{-3}$ | $3.390 \times 10^{-3}$ | $2.865 \times 10^{-3}$ | 73.63 |
| | 40,000 | $2.773 \times 10^{-4}$ | $7.568 \times 10^{-4}$ | $2.459 \times 10^{-3}$ | $2.305 \times 10^{-3}$ | 135.69 |

*5.2. Numerical Approximation of the Proposed Formulas with a Finite Sum*

According to Section 3.3, when $Q_j$ are non-zero for all $j \in \mathbb{N}_0$, the series (14) diverges. In this section, we study the level of accuracy of the formulas in Theorems 1 and Corol-

lary 3, when the values are estimated using their partial sum. We denote $U_E^{\langle \gamma, K \rangle}$ for the approximation of $U_E^\gamma$ by a partial sum of (4) up to order $\gamma - K$. In order to find a suitable number $K$, before comparing with MC simulations, we need to measure the significant difference of the value of $U_E^{\langle \gamma, K \rangle}$ at each $K \in \mathbb{N}$, which is defined by a sequence of absolute relative differences (ARDs),

$$D_E^{\langle \gamma, K \rangle}(x, \tau) := \left| \frac{U_E^{\langle \gamma, K \rangle}(x, \tau) - U_E^{\langle \gamma, K-1 \rangle}(x, \tau)}{U_E^{\langle \gamma, K \rangle}(x, \tau)} \right|,$$

for all $(x, \tau) \in (0, \infty) \times [0, \infty)$. Furthermore, the accuracy of $U_E^{\langle \gamma, K \rangle}(x, \tau)$ compared with MC for fixing the suitable $K$ is measured via the absolute relative errors (AREs) defined by

$$E_E^{\langle \gamma, K \rangle}(x, \tau) := \left| \frac{U_E^{\langle \gamma, K \rangle}(x, \tau) - U_E^{\langle \gamma, M \rangle}(x, \tau)}{U_E^{\langle \gamma, K \rangle}(x, \tau)} \right|,$$

for all $(R, \tau) \in (0, \infty) \times [0, \infty)$, where $U_E^{\langle \gamma, M \rangle}$ is the result of $\mathbb{E}^Q \left[ x_T^\gamma e^{- \int_t^T (\alpha x_s + \beta) \, ds} \mid x_t = x \right]$ received from MC simulations.

The sequences of ARDs of $D_E^{\langle \gamma \rangle}(x, 0.01)$ are displayed in Table 2 for $K = 5, 10, 15, 20$ with parameters $\theta = 1$, $\sigma_0 = \sigma_1 = 1$ and $d = 2$, except $x = 0.01, 1, 5$ for $\gamma = -1.5$, $-0.5, 0.5, 1.5$. In the case of infinite sum of $U_E^{\langle \gamma \rangle}(x, 0.01)$, we consider the parameters $\gamma = -1.5, -0.5, 0.5, 1.5$. According to Table 1, the received ARDs are likely improved when $K$ increases up to $K = 20$, showing that $U_E^{\langle \gamma, K \rangle}$ for these $K$ can already produce good approximations to $U_E^{\langle \gamma \rangle}$. To verify this claim, the results of $U_E^{\langle \gamma, K \rangle}$ with $K = 10$ are constructed to compare with MC simulations, and the results are shown in the next example.

**Table 2.** The ARDs $D_E^{\langle \gamma, K \rangle}(x, 0.01)$.

| x | K | $\gamma$ | | | |
|---|---|---|---|---|---|
| | | **−1.5** | **−0.5** | **0.5** | **1.5** |
| 0.1 | 5 | $2.605 \times 10^{-4}$ | $2.416 \times 10^{-6}$ | $2.985 \times 10^{-8}$ | $4.980 \times 10^{-9}$ |
| | 10 | $4.959 \times 10^{-6}$ | $1.262 \times 10^{-8}$ | $3.498 \times 10^{-11}$ | $9.897 \times 10^{-13}$ |
| | 15 | $8.822 \times 10^{-7}$ | $1.030 \times 10^{-9}$ | $1.226 \times 10^{-12}$ | $1.375 \times 10^{-14}$ |
| | 20 | $7.289 \times 10^{-7}$ | $4.866 \times 10^{-10}$ | $3.202 \times 10^{-13}$ | $1.912 \times 10^{-15}$ |
| 1 | 5 | $2.929 \times 10^{-9}$ | $2.445 \times 10^{-11}$ | $3.019 \times 10^{-13}$ | $5.490 \times 10^{-14}$ |
| | 10 | $5.576 \times 10^{-16}$ | $1.277 \times 10^{-18}$ | $3.539 \times 10^{-21}$ | $1.091 \times 10^{-22}$ |
| | 15 | $9.921 \times 10^{-22}$ | $1.043 \times 10^{-24}$ | $1.240 \times 10^{-27}$ | $1.515 \times 10^{-29}$ |
| | 20 | $8.196 \times 10^{-27}$ | $4.926 \times 10^{-30}$ | $3.238 \times 10^{-33}$ | $2.107 \times 10^{-35}$ |
| 5 | 5 | $9.460 \times 10^{-13}$ | $7.834 \times 10^{-15}$ | $9.672 \times 10^{-17}$ | $1.772 \times 10^{-17}$ |
| | 10 | $5.763 \times 10^{-23}$ | $1.309 \times 10^{-25}$ | $3.627 \times 10^{-28}$ | $1.127 \times 10^{-29}$ |
| | 15 | $3.281 \times 10^{-32}$ | $3.421 \times 10^{-35}$ | $4.068 \times 10^{-38}$ | $5.012 \times 10^{-40}$ |
| | 20 | $8.674 \times 10^{-41}$ | $5.170 \times 10^{-44}$ | $3.399 \times 10^{-47}$ | $2.230 \times 10^{-49}$ |

**Example 3.** *The partial sum of (4) up to order $\gamma - K$ with $\alpha = \beta = 1$ for $\gamma = -1.5, -0.5, 0.5, 1.5$.*

The comparison results between the formulas $U_E^{\langle \gamma, 10 \rangle}(x, 0.01)$ and MC simulations are shown in Table 3. MC simulations are performed by 5000, 10,000, 20,000, and 40,000 sample paths using 10,000 discretized steps. Table 3 shows the results of MC simulations that closely match with the approximate Formula (4) with better approximations (smaller AREs) when the number of sample paths increases. This confirms that the finite partial sum approximation of (4) is very accurate as compared by MC simulation.

**Table 3.** AREs $E_E^{\langle \gamma, 10 \rangle}(x, 0.01)$ of approximations $U_E^{\langle \gamma, 10 \rangle}(x, 0.01)$ and MC simulations.

| x | No. of Paths | $\gamma$ | | | |
|---|---|---|---|---|---|
| | | **−1.5** | **−0.5** | **0.5** | **1.5** |
| 0.1 | 5000 | $8.879 \times 10^{-3}$ | $1.724 \times 10^{-3}$ | $1.904 \times 10^{-3}$ | $5.297 \times 10^{-3}$ |
| | 10,000 | $4.499 \times 10^{-3}$ | $1.439 \times 10^{-3}$ | $1.209 \times 10^{-3}$ | $2.196 \times 10^{-3}$ |
| | 20,000 | $1.089 \times 10^{-3}$ | $6.156 \times 10^{-4}$ | $5.260 \times 10^{-4}$ | $1.532 \times 10^{-3}$ |
| | 40,000 | $9.658 \times 10^{-4}$ | $5.739 \times 10^{-4}$ | $1.281 \times 10^{-4}$ | $8.521 \times 10^{-4}$ |
| 1 | 5000 | $4.498 \times 10^{-3}$ | $7.423 \times 10^{-4}$ | $7.291 \times 10^{-4}$ | $2.107 \times 10^{-3}$ |
| | 10,000 | $1.681 \times 10^{-3}$ | $7.044 \times 10^{-4}$ | $6.195 \times 10^{-4}$ | $9.662 \times 10^{-4}$ |
| | 20,000 | $5.316 \times 10^{-4}$ | $5.866 \times 10^{-4}$ | $3.168 \times 10^{-4}$ | $9.356 \times 10^{-4}$ |
| | 40,000 | $1.468 \times 10^{-4}$ | $1.600 \times 10^{-4}$ | $2.496 \times 10^{-4}$ | $7.086 \times 10^{-4}$ |
| 5 | 5000 | $1.260 \times 10^{-3}$ | $1.531 \times 10^{-4}$ | $2.136 \times 10^{-4}$ | $4.115 \times 10^{-4}$ |
| | 10,000 | $8.906 \times 10^{-4}$ | $1.327 \times 10^{-4}$ | $1.785 \times 10^{-4}$ | $3.865 \times 10^{-4}$ |
| | 20,000 | $8.220 \times 10^{-4}$ | $7.713 \times 10^{-5}$ | $1.128 \times 10^{-4}$ | $1.216 \times 10^{-4}$ |
| | 40,000 | $9.445 \times 10^{-5}$ | $1.588 \times 10^{-5}$ | $3.812 \times 10^{-5}$ | $8.439 \times 10^{-5}$ |

## 6. Contingent Claims Pricing

In the context of pricing an option, assume that the underlying asset is set up to follow the ECIR process (3); we first define the following process

$$V_t := \mathbb{E}^Q \left[ e^{-\int_t^T (\alpha x_s + \beta)\, ds} f_T + \int_t^T e^{-\int_t^s (\alpha x_u + \beta)\, du} g_s \, ds \mid \mathcal{F}_t \right], \quad 0 \leq t \leq T, \qquad (25)$$

where $f_T$ and $g$ are nonnegative functions. In particular, according to Karatzas and Shreve's exercise 8.13 in [35], the process $V_t$ in (25) gives the unique wealth process with the initial wealth $x$; more details on this can be found in [35]. This is also called the valuation process of a contingent claim $(g, f_T)$, where $f_T$ is the terminal payoff at maturity and $g = \{g_t, \mathcal{F}_t \mid 0 \leq t \leq T\}$ is the payoff rate.

This section illustrates an application for valuing the contingent claim with a date of maturity $T$, which depends on the underlying asset $x_t$ following the ECIR or CIR process. The analytical formulas for a contingent claim are provided in the following theorems.

**Proposition 1.** *Let $x_t$ follow the ECIR process (3) with $\alpha, \beta \in \mathbb{R}$ and $n_1, n_2 \in \mathbb{N}_0$. Suppose that $f_T \equiv x_T^{n_1}$ and $g_s \equiv x_s^{n_2}$ for $0 \leq t \leq s \leq T$, then*

$$V_t = U_E^{n_1}(x, \tau) + \int_0^\tau U_E^{n_2}(x, v) \, dv, \qquad (26)$$

*for $(x, \tau) \in D_E \subset (0, \infty) \times [0, \infty)$, $\tau = T - t$, and $U_E^{n_1}$ and $U_E^{n_2}$ are given in Corollary 1.*

**Proof.** Applying Fubini's theorem and Corollary 1 yields

$$V_t = \mathbb{E}^Q \left[ x_T^{n_1} e^{-\int_t^T (\alpha x_s + \beta)\, ds} \mid x_t = x \right] + \int_t^T \mathbb{E}^Q \left[ x_s^{n_2} e^{-\int_t^s (\alpha x_u + \beta)\, du} \mid x_t = x \right] ds$$

$$= U_E^{n_1}(x, \tau) + \int_t^T U_E^{n_2}(x, s - t) \, ds.$$

Setting $v = s - t$ obtains (26) as required. $\square$

**Remark 5.** *Suppose that $f_T \equiv \sum_{k=0}^{n_1} a_k x_T^k$ and $g_s \equiv \sum_{k=0}^{n_2} b_k x_s^k$, where $x_t$ follows the ECIR process with $\alpha, \beta \in \mathbb{R}$ and $n_1, n_2 \in \mathbb{N}_0$, for some sequences of real numbers $(a_0, a_1, \dots, a_{n_1})$, $(b_0, b_1, \dots, b_{n_2})$ in which $a_{n_1}$ and $b_{n_2}$ are not zero. According to Proposition 1, we have*

$$
V_t = \mathbb{E}^Q\left[ e^{-\int_t^T (\alpha x_s + \beta)\, ds} \left( \sum_{k=0}^{n_1} a_k x_T^k \right) + \int_t^T e^{-\int_t^s (\alpha x_u + \beta)\, du} \left( \sum_{k=0}^{n_2} b_k x_s^k \right) ds \mid x_t = x \right]
$$

$$
= \sum_{k=0}^{n_1} a_k U_E^k(x, \tau) + \sum_{k=0}^{n_2} b_k \int_0^\tau U_E^k(x, v)\, dv.
$$

*Furthermore, for a CIR process, the above equation can readily be reduced to the following form*

$$
V_t = \sum_{k=0}^{n_1} a_k U_C^k(x, \tau) + \sum_{k=0}^{n_2} b_k \left( \sum_{j=0}^{k} \left( \int_0^\tau e^{B(v)x} A_j(v)\, dv \right) x^{k-j} \right).
$$

**Corollary 8.** *Suppose that $x_t$ follows the CIR process. According to Proposition 1 with $\alpha = 0$, $\beta = r > 0$ (also called fixed rate), and $n_1, n_2 \in \mathbb{N}_0$, we have*

$$
V_t = U_C^{n_1}(x, \tau) + \frac{1 - e^{-(r + n_2\theta)\tau}}{r + n_2\theta} x^{n_2}
$$

$$
+ \sum_{j=1}^{n_2} \left( \prod_{k=1}^{j} \frac{Q_k}{k\theta} \sum_{k=0}^{j} \left( (-1)^{j-k+1} \binom{j}{k} \frac{\left( e^{-(r + (n_2-k)\theta)\tau} - 1 \right)}{r + (n_2 - k)\theta} \right) \right) x^{n_2 - j}, \qquad (27)
$$

*for $(x, \tau) \in D_C \subset (0, \infty) \times [0, \infty)$, $\tau = T - t$, and $Q_k$ is given in Corollary 3.*

**Proof.** From (20) in Corollary 4 with $\alpha = 0$, then $\rho = \theta$, $B(\tau) = 0$, and $H_j(\tau) = e^{-(r + (\gamma_2 - j)\theta)\tau}$ for all $\tau \geq 0$ and $j \in \mathbb{N}_0$. Recalling Remark 5, we have

$$
V_t = U_C^{n_1}(x, \tau) + \sum_{j=0}^{n_2} \left( \int_0^\tau A_j(v)\, dv \right) x^{n_2 - j}.
$$

First, considering $\int_0^\tau A_j(v)\, dv$ for only $j = 0$ yields

$$
\int_0^\tau A_0(v)\, dv = \int_0^\tau H_0(v)\, dv = \frac{1 - e^{-(r + n_2\theta)\tau}}{r + n_2\theta},
$$

and the remaining terms, for $j = 1, 2, .., n_2$,

$$
\int_0^\tau A_j(v)\, dv = \int_0^\tau \left( \prod_{k=1}^{j} \frac{2Q_k}{k} \right) H_j(v) \left( \frac{e^{\theta v} - 1}{2\theta e^{\theta v}} \right)^j dv
$$

$$
= \prod_{k=1}^{j} \frac{Q_k}{k\theta} \int_0^\tau e^{-(r + (n_2 - j)\theta)v} \left( \frac{e^{\theta v} - 1}{e^{\theta v}} \right)^j dv
$$

$$
= \prod_{k=1}^{j} \frac{Q_k}{k\theta} \int_0^\tau e^{-(r + n_2\theta)v} (e^{\theta v} - 1)^j\, dv
$$

$$
= \prod_{k=1}^{j} \frac{Q_k}{k\theta} \sum_{k=0}^{j} (-1)^{j-k} \binom{j}{k} \int_0^\tau e^{-(r + (n_2 - k)\theta)v}\, dv
$$

$$
= \prod_{k=1}^{j} \frac{Q_k}{k\theta} \sum_{k=0}^{j} \left( (-1)^{j-k+1} \binom{j}{k} \frac{\left( e^{-(r + (n_2 - k)\theta)\tau} - 1 \right)}{r + (n_2 - k)\theta} \right). \qquad \square
$$

The benefits of these theorems to some well-known pricing instruments are shown in the following examples.

**Example 4.** *Zero-coupon bond.*

The valuation of a zero coupon bond at time $t$ with expiration date $T$, $p(t,T)$ is given by the expression

$$p(t,T) = \mathbb{E}^Q\left[e^{-\int_t^T \alpha x_s + \beta\, ds} \mid x_t = x\right]$$

where $x_t$ follows the ECIR process. Applying Corollary 1 by setting $\gamma = 0$, we obtain the formula for the price of the zero coupon bond

$$p(t,T) = A_0^{\langle 0 \rangle}(\tau)e^{B(\tau)x}.$$

In the case that $x_t$ follows the CIR process, Corollary 4 is used to produce the closed-form formula for valuing the zero coupon bond

$$p(t,T) = H_0(\tau)e^{B(\tau)x}$$

where

$$H_0(\tau) = \exp\left[\left(\frac{\theta^2\mu}{\sigma^2} - \beta + \frac{\theta\mu\rho}{\sigma^2}\right)\tau\right]\left(\frac{2\rho}{(\rho-\theta) + e^{\rho\tau}(\rho+\theta)}\right)^{\frac{2\theta\mu}{\sigma^2}},$$

$$B(\tau) = -\frac{2\alpha(e^{\rho\tau}-1)}{\rho(e^{\rho\tau}+1) + \theta(e^{\rho\tau}-1)},$$

and $\rho = \sqrt{\theta^2 + 2\alpha\sigma^2}$.

**Remark 6.** *If we set $\alpha = 1$ and $\beta = 0$ for the CIR process, we obtain the identical formula for the zero-coupon bond which appears in many pieces of literature.*

**Example 5.** *Two bonds interest rate swap.*

In this example, we apply the Corollary 1 for pricing the value of fixed rate for a floating swap, in which one company agrees to pay a fixed interest rate and receives in exchange a floating rate, see [21]. We consider the interest swap as the difference between the two bonds. From the point of view of the fixed ratepayer, the value of the interest rate swap, denoted by $P_{swap}$, is $P_{swap} := B_{float} - B_{fix}$ where $B_{float}$ is the value of floating rate bond, and $B_{fix}$ is the value of fixed rate bond; see [36] for more details.

Suppose that the value of the swap is zero at the initial time $t$ and the London Interbank Offered Rate (LIBOR), then zero rates are used as discount rates, denoted by $x_t$, which follows the ECIR process. Then

$$B_{fix} = \sum_{i=1}^{N} k_i e^{-\int_t^{T_i}(\alpha x_s + \beta)\, ds} + Le^{-\int_t^T(\alpha x_s + \beta)\, ds}$$

$$B_{float} = (L + k_0)e^{-\int_t^{T_1}(\alpha x_s + \beta)\, ds}$$

for some integer $N \geq 2$, where $t$ is the initial time, $T_i$ is the time until the $i$th payment is exchanged; $k_t$ is the fixed payment made at time $t$; $x_t$ is the LIBOR zero rates corresponding

to maturity $t$; and $L$ is the notional principal in swap agreement. Thus, the value of the interest rate swap at time $T$ is

$$\mathbb{E}^Q \left[ P_{swap} \mid x_t = x \right] \tag{28}$$

$$= \mathbb{E}^Q \left[ (L + k_0) e^{-\int_t^{T_1} (\alpha x_s + \beta)\, ds} - \left( \sum_{i=1}^N k_i e^{-\int_t^{T_i} (\alpha x_s + \beta)\, ds} + L e^{-\int_t^T (\alpha x_s + \beta)\, ds} \right) \mid x_t = x \right]$$

To calculate (28), Corollary 1 can be applied by setting $\gamma = 0$.

**Example 6.** *Arrears swap*

An arrears swap, also known as a delayed reset swap, is one of the traded instruments in the over-the-counter market, in which two companies or financial institutes decide to exchange periodic payments with another. In this interest rate fixed for floating swap, the floating rate paid on a payment date is based on the interest rate observed at the end of the reset period, as can be seen in [36] for more details.

Let $x_{fix}$ be a fixed rate, $x_t$ be a floating rate at time $t$, and $P$ be a notional principle. Suppose that an arrears swap has an expiration date $T$ with $N$ payment dates at $t = T_0 < T_1 < \cdots < T_N = T$ in an increment of $\Delta t = T_i - T_{i-1}$, $i = 1, 2, \ldots, N$. The payoff of such a swap from a floating rate payer's point of view at the $i^{\text{th}}$ payment date, $V_i^{ar}$, is the difference between interest in a notional principle considered by the fixed and floating interest rates, which can be expressed in the form $V_i^{ar} = \left( x_{fix} - x_{T_i} \right) \Delta t P$. By the fundamental theorem of asset pricing [3], a no-arbitrage price at any time $t$ of the arrears swap, $V^{\text{ar}}$, is the conditional expectation of the sum of each payoff discounted to the initial time $t = 0$, which is

$$V^{ar} = \mathbb{E}^Q \left[ \sum_{i=1}^N V_i^{ar} e^{-\int_t^{T_i} \alpha x_s + \beta\, ds} \mid x_t = x \right]$$

$$= \Delta t P \left( x_{fix} \sum_{i=1}^N \mathbb{E}^Q \left[ e^{-\int_t^{T_i} \alpha x_s + \beta\, ds} \mid x_t = x \right] - \sum_{i=1}^N \mathbb{E}^Q \left[ x_{T_i} e^{-\int_t^{T_i} \alpha x_s + \beta\, ds} \mid x_t = x \right] \right). \tag{29}$$

By applying Corollary 1 and setting $\gamma = 0$ and $\gamma = 1$, the value of the arrears swap (29) can be obtained as an analytical form. It should be noted that the fair value for paying the fixed rate is

$$x_{fix} = \frac{\sum_{i=1}^N \mathbb{E}^Q \left[ x_{T_i} e^{-\int_t^{T_i} \alpha x_s + \beta\, ds} \mid x_t = x \right]}{\sum_{i=1}^N \mathbb{E}^Q \left[ e^{-\int_t^{T_i} \alpha x_s + \beta\, ds} \mid x_t = x \right]} = \frac{\sum_{i=1}^N U_E^1(x, i\Delta t)}{\sum_{i=1}^N U_E^0(x, i\Delta t)}.$$

## 7. Conclusions

In this work, we proposed the analytical formula for a contingent claim with the polynomial payoff under the ECIR process represented as the conditional expectation of the product of polynomial and exponential functions, $\mathbb{E}^Q \left[ x_T^\gamma e^{-\int_t^T (\alpha x_s + \beta)\, ds} \mid x_t = x \right]$ where $\alpha, \beta, \gamma \in \mathbb{R}$ and $x_t$ follow the ECIR process (3). By solving (8) from the Feynman–Kac representation, the analytical formula of (2) for the ECIR process was constructed in terms of an infinite sum of analytical expressions in Theorem 1. Interestingly, the infinite sum is reduced to a finite sum if $\gamma \in \mathbb{N}_0$ in Corollaries 1 and 2. Under the CIR process, the parameters in (5) can be evaluated and the Riccati differential Equation (6) can be analytically solved; thus, the formula in Theorem 1 for (2) can be rapidly deduced to the formula in Corollary 3. In addition, as shown in Corollaries 4 and 5, the formulas can be stated in closed form. The formula for the unconditional expectations are also obtained by taking $T \to \infty$, as can be seen in Corollaries 6 and 7.

The proposed closed-form formulas validated the validity and efficiency by comparing the results with Monte Carlo simulations as illustrated in Section 5. The applications of the

proposed formulas for contingent claims are described in Section 6 for financial products such as zero-coupon bond, two bonds interest rate swap, an arrears swap. For these applications under the ECIR process (3), the formula of (2) is extended for a more general form, $\mathbb{E}^Q\left[e^{-\int_t^T (\alpha x_s + \beta)\,ds} f_T + \int_t^T e^{-\int_t^s (\alpha x_u + \beta)\,du} g_s\,ds \mid \mathcal{F}_t\right]$, for some real value functions $f_T$ and $g_s$ where $0 \le t < s \le T$. The proposed formulas can also be considered as generalized results of other formulas which appeared in the literature.

**Author Contributions:** Conceptualization, F.N. and K.M.; methodology, F.N. and K.M.; software, F.N.; validation, K.M.; formal analysis, K.M.; investigation, F.N.; writing—original draft preparation, F.N.; writing—review and editing, K.M.; visualization, F.N.; supervision, K.M.; project administration, K.M. All authors have read and agreed to the published version of the manuscript.

**Funding:** This research received no external funding.

**Institutional Review Board Statement:** Not applicable.

**Informed Consent Statement:** Not applicable.

**Data Availability Statement:** Not applicable.

**Acknowledgments:** Authors thank the Development and Promotion of Science and Technology Talents (DPST), which is a project of the Institute for the Promotion of Teaching Science and Technology (IPST) for their kind support.

**Conflicts of Interest:** The authors declare no conflict of interest.

## Abbreviations

The following abbreviations are used in this manuscript:

| | |
|---|---|
| ARD | absolute relative difference |
| ART | average run time |
| CIR | Cox–Ingersoll–Ross |
| ECIR | extended Cox–Ingersoll–Ross |
| EM | Euler–Maruyama |
| FRA | forward rate agreement |
| IRS | interest rate swap |
| LIBOR | London Interbank Offered Rate |
| MAD | mean absolute difference |
| MC | Monte Carlo |
| ODE | ordinary differential equation |
| OU | Ornstein–Uhlenbeck |
| PDE | partial differential equation |
| PDF | probability density function |
| RK4 | fourth-order Runge–Kutta |
| SDE | stochastic differential equation |

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
