# Peer review of "Analytically Pricing Formula for Contingent Claim with Polynomial Payoff under ECIR Process"

_symmetry, doi:10.3390/sym14050933_

Round 1

Author Response

Dear Reviewer,
The four suggestions have all been followed, namely,
1. Page 3, line 117, the authors mentioned that the proposed analytical formula
in [2] is under γ = α = β = 0, whereas in this case we have the expectation of
the function valued one anywhere and this expectation would have been 1.
— We checked and agreed with the comment. This part has been deleted to avoid confusion.
2. Page 6, line 164, the author mentioned that ”the function B is a generalized
Riccati’s equation”, whereas, the function B is the solution of a generalized
Riccati’s equation.
— We have checked the mistake as suggested and made a correction to this error.
3. Page 9, Equation (22) is a limit function as T tends to 1 and not be depend on
T after getting the limit, therefore, the notation  is not understandable.
They should change the notation and correct also it in the end of the proof
of Theorem 8.
— We agree with the comment. The notation  has been changed (See pdf.)
4. Page 10, Section 3.3, the term ”the series (14) converge”, in two place, should
be ”the series (14) converges”.
— They have been corrected.   Sincerely yours, Khamron Mekchay  

Reviewer 2 Report

Dear Authors

The paper investigate the different kinds of contingent claims and obtained analytical solution using the financial concepts such as  Feynman–Kac formula Theorem under the ECIR process. Also in some cases it consider some numerical methods such Euler method as well integration numerical techniques such as Runke Kutta methods. At final it compares the obtained results with the classical MC simulation which they are agree with each other's.  

The paper is well written and I support the publication of the paper, but also mention some minor revisions: 

  1. Please also cite the run time in tables as the time consumption is very important for the employed algorithm.
  2. The MC simulation is done based on Euler–Maruyama method which is in order of (O(h^1/2 ), So, I propose to test some higher order SDE numerical methods such as Mileten method.
  3. Please make the following papers in the body of the paper as well in the reference section:
    • D. Ahmadian, L. V. Ballestra, The Finite Element Method: A High-Performing Approach for Computing the Probability of Ruin and Solving Other Ruin-Related Problems, Mathematical Methods in the Applied Sciences, Vol. 44, Nov. 2021.
    • D. Ahmadian, L. V. Ballestra, Pricing geometric Asian rainbow options under the mixed fractional Brownian motion, Physica A: Statistical Mechanics and its Applications, Vol. 555, Oct. 2020.  

Author Response

Dear Reviewer,

The three suggestions have all been followed, namely,

  1. Please also cite the run time in tables as the time consumption is very important for the employed algorithm.
    — As suggested, we have included the run time of the MC simulations in table 1 compared to that from the formula (see Sec.5 pages: 13-14).

  2. The MC simulation is done based on Euler–Maruyama method which is in order of O(h1/2), So, I propose to test some higher order SDE numerical methods such as Milstein method.
    — In our work, we only use Euler-Maruyama method as MC simulation to validate the closed-form formula. Even though, this method only converges with order 12, the simulation results agree well with the results form the proposed formula.

  3. Please make the following papers in the body of the paper as well in the reference section:

    • D. Ahmadian, L. V. Ballestra, The Finite Element Method: A High-Performing Approach for Computing the Probability of Ruin and Solving Other Ruin-Related Problems, Mathematical Methods in the Applied Sciences, Vol. 44, Nov. 2021.

    • D. Ahmadian, L. V. Ballestra, Pricing geometric Asian rainbow options under the mixed fractional Brownian motion, Physica A: Statistical Mechanics and its Applications, Vol. 555, Oct. 2020.

      — We have revised the manuscript to include details related to the suggested works (see Sec 1 page: 2 .)

      • We agree with the comment on English language. We will send the manuscript for language editing service suggested by the MDPI, once it is accepted.
      Sincerely yours,
    • Khamron Mekchay

Reviewer 3 Report

See the enclosed report

Author Response

Dear Reviewer,

The three suggestions have all been followed, namely,

  1. Page 1, Line −1: Coefficient γ is said to be in R before being introduced in a formula; it does not appear in formula (1). Coefficient γ is introduced in formula (2) in page 2.
    — We have revised as suggested.

  2. In my opinion, Theorems from 2 to 8, despite being very interesting since the point of view of applications, are quite straightforward corollaries of Theorem 1. I would name it as corollaries. Theorem 9 is also a corollary of Theorem 1, but with more entity, so I would name it as a proposition.
    — We agree with the comment. The Theorem from 2 to 8 and Theorem 9 have been changed to Corollary 1 to 7 and Proposition 1, respectively.

  3. The paper is very well presented and nicely written, but I guess the English style could be improved.
    — We agree with the comment and will send the manuscript for language editing service suggested by the MDPI, once it is accepted.

    Sincerely yours,
    Khamron Mekchay
